# The influence of pneumococcal positivity on clinical outcomes among patients hospitalized with COVID-19: A retrospective cohort study

Chang-Seok Yoon [1], Ha-Young Park[1], Hwa-Kyung Park[1], Jae-Kyeong Lee[1], Bo-Gun Kho[1,2], Hong-Joon Shin[1,2], Yong-Soo Kwon[1,2], Yu-Il Kim[1,2], Sung-Chul Lim[1,2], Tae-Ok Kim [1,2]*

1 Department of Internal Medicine, Division of Pulmonology and Critical Care Medicine, Chonnam National University Hospital, Gwangju, South Korea, 2 Department of Internal Medicine, Chonnam National University Medical School, Gwangju, South Korea

* tokim@jnu.ac.kr

## Abstract

### Background

Bacterial co-infection has been associated with adverse outcomes in patients with COVID-19. *Streptococcus pneumoniae* is a common cause of community-acquired pneumonia and may contribute to poor clinical outcomes when co-detected in COVID-19 patients. This study aimed to investigate the clinical significance of pneumococcal positivity in hospitalized patients with COVID-19.

### Methods

We conducted a retrospective analysis of adult patients hospitalized with COVID-19 at two tertiary care centers. Pneumococcal positivity was defined by either a positive urinary antigen test or multiplex real-time polymerase chain reaction. Disease severity of COVID-19 pneumonia was assessed using the pneumonia severity index and CURB-65 scoring systems. Propensity score matching and multivariable logistic regression were used to adjust for confounders and identify independent risk factors for mortality.

### Results

Among 280 patients, 65 pneumococcus-positive patients were matched with 65 pneumococcus-negative patients after propensity score matching. In the overall matched cohort, pneumococcal positivity was not significantly associated with in-hospital mortality. However, in patients with severe disease (n = 156), defined as pneumonia severity index >130 or CURB-65 ≥ 3, mortality was significantly higher in pneumococcus-positive patients (n = 39) than in pneumococcus-negative patients (53.8% vs. 29.1%, $p = 0.009$). In the multivariable analysis of this subgroup,

**Data availability statement:** The raw data have been provided as Supporting Information.

**Funding:** This research was supported by the National Research Foundation of Korea (2022R1F1A1069623) and the Chonnam National University Hospital Biomedical Research Institute (BCRI22033).

**Competing interests:** The authors have declared that no competing interests exist.

pneumococcal positivity (odds ratio, 4.050; 95% confidence interval, 1.285–12.765; $p = 0.017$) and high-flow oxygen therapy (odds ratio, 6.510; 95% confidence interval, 1.847–22.944; $p = 0.004$) were independently associated with mortality.

## Conclusion

Detection of *S. pneumoniae* by urinary antigen test or multiplex polymerase chain reaction was associated with increased mortality in patients hospitalized with severe COVID-19.

## Introduction

COVID-19, caused by SARS-CoV-2, was declared a global pandemic in March 2020. Since its emergence, over 65.8 million cases have been reported worldwide, with mortality exceeding 1.5 million [1]. The clinical presentation of COVID-19 varies widely, ranging from asymptomatic or mild upper respiratory symptoms to severe pneumonia, acute respiratory distress syndrome, multi-organ failure, and death. Several prognostic factors have been associated with disease severity and poor clinical outcomes in patients with COVID-19. Advanced age, male sex, and underlying comorbidities such as hypertension, diabetes mellitus, cardiovascular disease, chronic lung disease, obesity, and immunosuppression are significant independent factors [2].

Bacterial co-infections have also been recognized as factors contributing to adverse clinical outcomes in COVID-19 patients. Such co-infections are associated with higher mortality rates, a greater likelihood of intensive care unit (ICU) admission, and an increased need for invasive mechanical ventilation (IMV) [3,4]. Common bacterial pathogens identified in COVID-19 patients include *Streptococcus pneumoniae*, *Staphylococcus aureus, Haemophilus influenzae*, and *Klebsiella pneumoniae* [5].

*Streptococcus pneumoniae*, one of the most common pathogens in community-acquired pneumonia (CAP), is known to cause lower respiratory tract infections through interactions with respiratory viruses, including SARS-CoV-2 [6,7]. To facilitate early and accurate identification of *Streptococcus pneumoniae*, diagnostic methods such as urinary antigen tests (UATs) and multiplex polymerase chain reaction (PCR) assays are frequently used. However, these assays may demonstrate relatively low specificity, particularly due to their inability to differentiate colonization from true infection. Despite these limitations, both UAT and multiplex PCR have become important diagnostic tools in routine clinical practice, aiding in the early identification of co-infections and guiding the timely initiation of appropriate antimicrobial therapy.

In this context, we aimed to evaluate whether *Streptococcus pneumoniae* positivity is associated with adverse clinical outcomes by comparing COVID-19 patients with and without detection of *Streptococcus pneumoniae*.

## Materials and methods

### Population

This retrospective study was conducted at Chonnam National University Hospital and Chonnam National University Hwasun Hospital from January 2020 to June

2023. Inclusion criteria were as follows: patients who were [1] older than 18 years, [2] diagnosed with COVID-19 via rapid antigen test or real-time reverse transcription PCR test, and [3] underwent streptococcus UAT and multiplex PCR testing for respiratory pathogens at the time of COVID-19 diagnosis. Patients who were diagnosed with COVID-19 but did not require hospitalization, as well as those diagnosed with COVID-19 during hospitalization for unrelated medical conditions, were excluded from the analysis.

To analyze the impact of pneumococcal positivity on in-hospital outcomes among patients with COVID-19, we collected demographic and clinical data, including age, sex, and underlying medical conditions. Laboratory parameters such as white blood cell count, neutrophil count, lymphocyte count, C-reactive protein, and procalcitonin levels were also recorded. Pneumonia severity was assessed using validated scoring systems, including the pneumonia severity index (PSI) and the CURB-65 score. Details of administered treatments, including supplemental oxygen, organ support, antibiotic therapy, and antiviral agents, were documented. The outcome measures analyzed included in-hospital mortality, length of hospital stay, and duration of ICU admission.

### Definition of pneumococcus-positive

The term "pneumococcus-positive" was defined as fulfillment of at least one of the following criteria:

1. A positive result from multiplex real-time PCR using the Allplex PneumoBacter Assay (Seegene, Seoul, Korea) or

2. A positive result from a rapid UAT for *S. pneumoniae* using the BinaxNOW assay (Binax, Portland, ME, USA).

### Definition of severe COVID-19 pneumonia

Severe COVID-19 pneumonia was defined based on the PSI and CURB-65 scoring systems. Patients with a PSI greater than 130 or a CURB-65 score greater than 3 were categorized as having severe disease.

### Statistical analysis

All data were expressed as means and standard deviations or as numbers (%) in the text and tables. Intergroup comparisons were performed using the independent t-test for continuous variables following a normal distribution, the Wilcoxon rank-sum test for continuous variables not following a normal distribution, and Pearson's $\chi^2$ or Fisher's exact tests for categorical variables. To identify risk factors associated with mortality, logistic regression analysis was conducted. Univariable logistic regression was first performed using demographic characteristics, comorbidities, and clinical variables. Variables with a $p$-value < 0.2 in the univariable analysis were subsequently included in a multivariable logistic regression model using the backward selection method. Throughout the analysis, a $p$-value < 0.05 was considered statistically significant.

Propensity score matching (PSM) was applied in the comparative analysis to adjust for confounders and rigorously evaluate the impact of pneumococcal positivity on clinical outcomes. The matching process included demographic variables, such as age, sex, and comorbidities, as well as severity scores like the PSI and CURB-65. Additionally, initial treatments—including the use of antiviral agents, corticosteroids, antibiotics, and oxygen delivery methods—were incorporated into the matching procedure. PSM was performed using the nearest-neighbor approach with a 1:1 matching ratio. To maintain the matching ratio and ensure inclusion of all treated patients, the caliper width was set to 0.2 standard deviations of the logit of the propensity score. To confirm the appropriateness of the matching, we evaluated the balance of matched variables between the two groups using standardized mean differences (SMDs) as the primary measure. Overall, covariate balance was relatively well achieved through propensity score matching, with the majority of variables showing SMDs below 0.1, although some imbalance remained in a few variables (S1 Table). In addition, t-tests and chi-square tests were also performed, where appropriate, to provide supplementary comparisons. All analyses were conducted using SPSS Statistics for Windows, version 26.0 (IBM Corp., Armonk, NY, USA).

## Ethics statement

The study was approved by the Chonnam National University Hospital Institutional Review Board (IRB approval number: CNUH-2024–025) and was performed in accordance with the principles of the Declaration of Helsinki. The requirement for informed consent was waived by the Chonnam National University Hospital Institutional Review Board due to the retrospective nature of the study. The clinical data were accessed for research purposes on 1 February 2024. All data were fully anonymized before analysis, and the investigators did not have access to any personally identifiable information during or after data collection.

## Results

In total, 507 patients with COVID-19 underwent a UAT or sputum multiplex PCR for *Streptococcus pneumoniae* during the study period (Fig 1). Among these patients, 280 were ultimately included in the study, after the exclusion of 42 patients younger than 18 years, 40 patients who were not admitted to the hospital, 102 patients diagnosed with COVID-19 during hospitalization, and 43 patients without pneumonia. Of these included patients, 65 tested positive for *Streptococcus pneumoniae* by UAT or multiplex PCR, while 215 tested negative.

### Comparisons of basal characteristics between pneumococcus-positive group and pneumococcus-negative group

Table 1 presents a detailed comparison of clinical characteristics and outcomes between pneumococcus-positive and pneumococcus-negative COVID-19 patients. In the pneumococcus-positive group, the mean age was 76.14 ± 13.38 years, and 63.1% were male (n = 41). Overall, baseline demographic characteristics were not significantly different between the two groups. However, hypertension was more prevalent in the pneumococcus-positive group than in the pneumococcus-negative group (61.5% vs. 46.5%, *p* = 0.047). In contrast, hematologic malignancies were observed more frequently in the pneumococcus-negative group than in the pneumococcus-positive group (17.7% vs. 4.6%, *p* = 0.008). High-flow oxygen

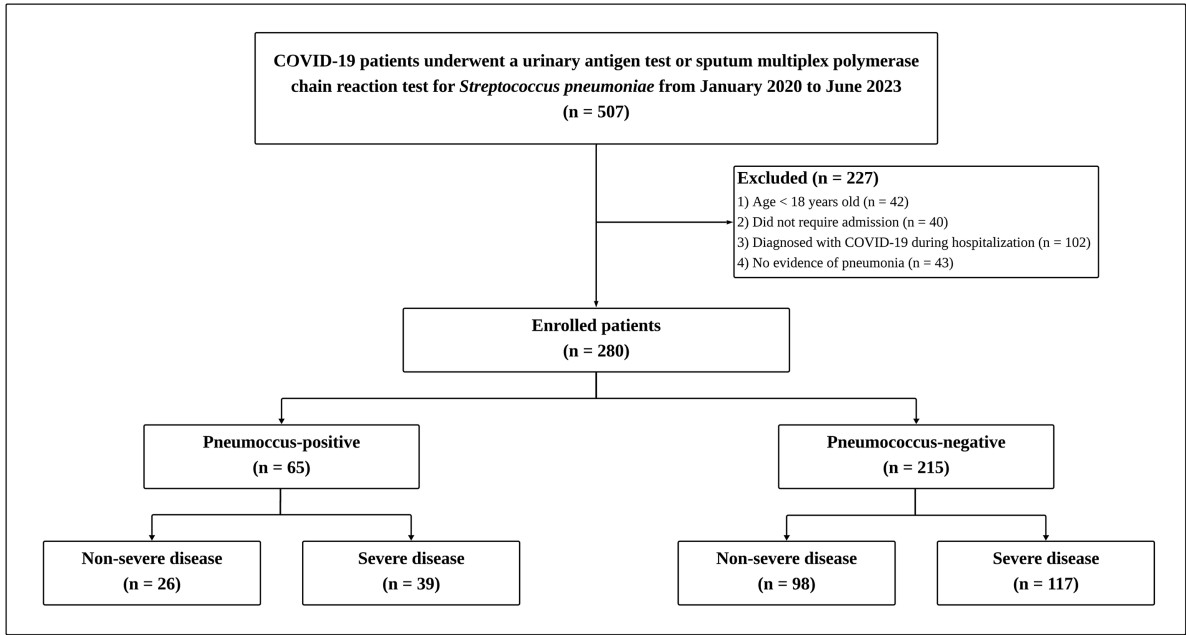

**Fig 1. Study enrollment flow chart.**

therapy (HFOT) was administered more frequently to patients in the pneumococcus-positive group (50.8% vs. 35.3%, $p = 0.03$), whereas no significant differences were observed between groups regarding the use of non-invasive ventilation (NIV) or IMV. Among laboratory results, procalcitonin levels were significantly higher in patients who tested positive for pneumococcus compared to those who tested negative (13.98 vs. 4.00 ng/mL, $p = 0.007$). Similarly, patients in the

**Table 1. Characteristics and clinical outcomes in patients with COVID-19 with and without pneumococcal positivity.**

| Variables | Without propensity score matching | | | With propensity score matching | | |
|---|---|---|---|---|---|---|
| | Pneumococcus-negative (n = 215) | Pneumococcus-positive (n = 65) | p-value | Pneumococcus negative (n = 65) | Pneumococcus positive (n = 65) | p-value |
| Sex, male (%) | 142 (66.0) | 41 (63.1) | 0.658 | 45 (69.2) | 41 (63.1) | 0.458 |
| Age, years, mean ± SD | 73.16 ± 13.75 | 76.14 ± 13.38 | 0.121 | 75.97 ± 11.74 | 76.14 ± 13.38 | 0.939 |
| Underlying diseases | | | | | | |
| Hypertension, n (%) | 100 (46.5) | 40 (61.5) | 0.047 | 37 (56.9) | 40 (61.5) | 0.592 |
| Diabetes mellitus, n (%) | 76 (35.3) | 29 (44.6) | 0.190 | 31 (47.7) | 29 (44.6) | 0.725 |
| Chronic lung disease, n (%) | 73 (34.0) | 21 (32.3) | 0.881 | 20 (30.8) | 21 (32.3) | 0.850 |
| Chronic heart disease, n (%) | 56 (26.0) | 25 (38.5) | 0.062 | 29 (44.6) | 25 (38.5) | 0.477 |
| Chronic kidney disease, n (%) | 50 (23.3) | 16 (24.6) | 0.868 | 23 (35.4) | 16 (24.6) | 0.251 |
| Chronic liver disease, n (%) | 8 (3.7) | 1 (1.5) | 0.382 | 2 (3.1) | 1 (1.5) | 0.559 |
| Solid cancer, n (%) | 52 (24.2) | 11 (16.9) | 0.240 | 11 (16.9) | 11 (16.9) | >0.999 |
| Hematologic disease, n (%) | 38 (17.7) | 3 (4.6) | 0.008 | 1 (1.5) | 3 (4.6) | 0.310 |
| Cerebrovascular disease, n (%) | 50 (23.3) | 19 (29.2) | 0.329 | 18 (27.7) | 19 (29.2) | 0.846 |
| Oxygen therapy | | | | | | |
| High-flow oxygen therapy, n (%) | 76 (35.3) | 33 (50.8) | 0.030 | 33 (50.8) | 33 (50.8) | >0.999 |
| Non-invasive ventilation, n (%) | 1 (0.5) | 2 (3.1) | 0.136 | 0 (0.0) | 2 (3.1) | 0.156 |
| Mechanical ventilation, n (%) | 43 (20.0) | 11 (16.9) | 0.720 | 10 (15.4) | 11 (16.9) | 0.812 |
| Laboratory findings | | | | | | |
| WBC, x10³/μl, mean ± SD | 8.90 ± 6.24 | 8.85 ± 5.60 | 0.957 | 9.13 ± 5.31 | 8.86 ± 5.60 | 0.777 |
| Neutrophil, x10³/μl, mean ± SD | 7.28 ± 5.70 | 7.41 ± 5.37 | 0.870 | 7.61 ± 5.16 | 7.41 ± 5.37 | 0.825 |
| Lymphocyte, x10³/μl, mean ± SD | 1.02 ± 1.43 | 0.98 ± 0.64 | 0.746 | 0.89 ± 0.63 | 0.98 ± 0.64 | 0.396 |
| CRP, mg/dL, mean ± SD | 12.01 ± 8.94 | 12.13 ± 9.96 | 0.932 | 13.36 ± 8.91 | 12.12 ± 9.96 | 0.459 |
| Procalcitonin, ng/mL, mean ± SD | 4.00 ± 16.19 | 13.98 ± 41.30 | 0.007 | 6.32 ± 26.16 | 12.26 ± 38.91 | 0.310 |
| Severity scores | | | | | | |
| CURB-65, mean ± SD | 2.03 ± 1.22 | 2.45 ± 1.29 | 0.022 | 2.34 ± 1.09 | 2.45 ± 1.29 | 0.608 |
| PSI, mean ± SD | 130.90 ± 43.15 | 141.40 ± 46.90 | 0.113 | 137.37 ± 41.27 | 141.35 ± 46.97 | 0.608 |
| Treatment | | | | | | |
| Antiviral agents[h], n (%) | 183 (85.1) | 59 (90.8) | 0.244 | 56 (86.2) | 57 (87.7) | 0.795 |
| Corticosteroids[‡], n (%) | 147 (68.4) | 48 (73.8) | 0.400 | 42 (64.6) | 48 (73.8) | 0.342 |
| Antibiotics[§], n (%) | 167 (77.7) | 50 (76.9) | 0.899 | 52 (80.0) | 49 (75.4) | 0.527 |
| Clinical outcomes | | | | | | |
| Length of hospital stay, days, mean ± SD | 14.93 ± 13.40 | 13.62 ± 13.27 | 0.485 | 13.29 ± 10.83 | 13.62 ± 13.27 | 0.879 |
| Mortality, n (%) | 41 (19.1) | 22 (33.8) | 0.017 | 13 (20.0) | 22 (33.8) | 0.075 |
| Secondary bacteremia, n (%) | 19 (8.8) | 11 (16.9) | 0.140 | 9 (13.9) | 11 (16.9) | 0.627 |

Abbreviations: SD, standard deviation; WBC, white blood cells; CRP, C-reactive protein; PSI, pneumonia severity index

[h] Antiviral agents such as remdesivir, molnupiravir, nirmatrelvir/ritonavir, and others used in the treatment of COVID-19.

[‡] Refers to all corticosteroids, including dexamethasone and methylprednisolone, used in the treatment of COVID-19.

[§] Refers to empirical or targeted antibiotic therapy for confirmed or suspected bacterial infection.

pneumococcus-positive group had significantly higher CURB-65 scores (2.45 vs. 2.03, $p = 0.022$). However, no significant difference was noted in PSI between the two groups. Regarding clinical outcomes, the in-hospital mortality rate was significantly higher among pneumococcus-positive group compared to pneumococcus-negative group (33.8% vs. 19.1%, $p = 0.017$). There were no significant differences between the two groups in terms of secondary bacteremia or length of hospital stay.

Among the 280 patients included in the analysis, 156 (55.7%) were classified as having severe disease, defined as a PSI > 130 or a CURB-65 score≥3. Of these severely ill patients, 39 (25.0%) belonged to the pneumococcus-positive group. The overall mortality rate among patients with severe disease was 35.3%. Table 2 presents a detailed comparison of clinical characteristics and outcomes between pneumococcus-positive and pneumococcus-negative COVID-19 patients. Among severe COVID-19 patients, those positive for pneumococcus had a higher prevalence of hypertension (74.4% vs. 53.8%, $p = 0.024$) and chronic heart disease (51.3% vs. 31.6%, $p = 0.027$) but a lower prevalence of solid cancer (12.8% vs. 29.1%, $p = 0.043$). HFOT was also administered more often in the pneumococcus-positive group (66.7% vs. 44.4%, $p = 0.016$). However, there was no significant difference between the groups in the use of NIV or IMV. The administration of therapeutic agents—including antivirals, corticosteroids, and antibiotics—was comparable between the groups. Patients in the pneumococcus-positive group had higher CURB-65 scores than those in the pneumococcus-negative group (3.23 vs. 2.79, $p = 0.014$). In contrast, no significant difference was observed in PSI between the two groups. Notably, patient mortality was significantly higher in the pneumococcus-positive group compared to the pneumococcus-negative group (53.8% vs. 29.1%, $p = 0.009$).

### Relationship between pneumococcal positivity and mortality in COVID-19 patients

In the multivariable analysis (Table 3), the PSI (odds ratio [OR], 1.028; 95% confidence interval [CI], 1.017–1.039; $p < 0.001$), HFOT (OR, 6.524; 95% CI, 2.892–14.718; $p < 0.001$), and IMV (OR, 9.994; 95% CI, 4.423–22.583; $p < 0.001$) were independently associated with in-hospital mortality. However, pneumococcal positivity was not significantly associated with mortality in the overall cohort.

PSM was performed to adjust for baseline differences between pneumococcus-positive and pneumococcus-negative COVID-19 patients. After matching, 65 pairs of patients were identified. In the matched cohort, there was no significant difference in in-hospital mortality between pneumococcus-positive and pneumococcus-negative groups (Table 1). In the multivariable analysis of the matched cohort, PSI (OR, 1.025; 95% CI, 1.010–1.039; $p = 0.001$), HFOT (OR, 9.173; 95% CI, 2.720–30.933; $p < 0.001$), and IMV (OR, 3.601; 95% CI, 1.084–11.963; $p = 0.036$) were significantly associated with mortality (Table 3).

### Relationship between pneumococcal positivity and mortality in severe COVID-19 patients

In the multivariable analysis limited to patients with severe disease (Table 4), the following variables were independently associated with mortality: PSI (OR, 1.031; 95% CI, 1.015–1.047; $p < 0.001$), HFOT (OR, 4.530; 95% CI, 1.812–11.330; $p = 0.001$), IMV (OR, 7.537; 95% CI, 2.961–19.186; $p < 0.001$), and hematologic malignancies (OR, 4.998; 95% CI, 1.456–17.157; $p = 0.011$). Notably, pneumococcal positivity was also independently associated with mortality in this subgroup (OR, 3.286; 95% CI, 1.299–8.308; $p = 0.012$).

We also conducted a subgroup analysis for severe disease among matched patients. In this subgroup analysis, 39 pairs of pneumococcus-positive and pneumococcus-negative patients were matched. Among matched patients with severe COVID-19, the pneumococcus-positive group had a significantly higher mortality rate compared to the pneumococcus-negative group (53.8% vs. 23.1%; $p = 0.005$) (Table 2). In the multivariable analysis, HFOT (OR, 6.510; 95% CI, 1.847–22.944; $p = 0.004$), PSI score (OR, 1.028; 95% CI, 1.005–1.051; $p = 0.018$), and pneumococcal positivity (OR, 4.050; 95% CI, 1.285–12.765; $p = 0.017$) were independently associated with in-hospital mortality (Table 4).

**Table 2. Characteristics and clinical outcomes of patients with severe COVID-19, with and without pneumococcal positivity.**

| Variables | Without propensity score matching | | | With propensity score matching | | |
|---|---|---|---|---|---|---|
| | Pneumococcus negative n=117 | Pneumococcus positive n=39 | *p*-value | Pneumococcus negative n=39 | Pneumococcus positive n=39 | *p*-value |
| Sex, male (%) | 86 (73.5) | 27 (69.2) | 0.605 | 30 (76.9) | 27 (69.2) | 0.444 |
| Age, years, mean±SD | 77.62±9.29 | 80.56±8.13 | 0.079 | 78.49±7.88 | 80.56±8.13 | 0.255 |
| Underlying diseases | | | | | | |
| Hypertension, n (%) | 63 (53.8) | 29 (74.4) | 0.024 | 25 (64.1) | 29 (74.4) | 0.326 |
| Diabetes mellitus, n (%) | 51 (43.6) | 20 (51.3) | 0.403 | 24 (61.5) | 20 (51.3) | 0.361 |
| Chronic lung disease, n (%) | 41 (35.0) | 13 (33.3) | 0.846 | 12 (30.8) | 13 (33.3) | 0.808 |
| Chronic heart disease, n (%) | 37 (31.6) | 20 (51.3) | 0.027 | 22 (56.4) | 20 (51.3) | 0.650 |
| Chronic kidney disease, n (%) | 38 (32.5) | 13 (33.3) | 0.922 | 17 (43.6) | 13 (33.3) | 0.352 |
| Chronic liver disease, n (%) | 5 (4.30) | 1 (2.60) | 0.631 | 2 (5.1) | 1 (2.6) | 0.556 |
| Solid cancer, n (%) | 34 (29.1) | 5 (12.8) | 0.043 | 8 (20.5) | 5 (12.8) | 0.362 |
| Hematologic disease, n (%) | 16 (13.7) | 1 (2.6) | 0.054 | 0 (0.0) | 1 (2.6) | 0.314 |
| Cerebrovascular disease, n (%) | 39 (33.3) | 17 (43.6) | 0.248 | 14 (35.9) | 17 (43.6) | 0.488 |
| Oxygen therapy | | | | | | |
| High flow oxygen therapy, n (%) | 52 (44.4) | 26 (66.7) | 0.016 | 20 (51.3) | 26 (66.7) | 0.250 |
| Non-invasive mechanical ventilation, n (%) | 0 (0.0) | 0 (0.0) | >0.999 | 0 (0.0) | 0 (0.0) | >0.999 |
| Invasive mechanical ventilation, n (%) | 36 (30.8) | 10 (25.6) | 0.543 | 7 (17.9) | 10 (25.6) | 0.411 |
| Laboratory findings | | | | | | |
| WBC, x10³/µl, mean±SD | 10.06±5.98 | 9.79±6.47 | 0.814 | 10.22±5.58 | 9.79±6.47 | 0.754 |
| Neutrophil, x10³/µl, mean±SD | 8.58±5.74 | 8.56±6.20 | 0.981 | 8.75±5.39 | 8.56±6.20 | 0.887 |
| Lymphocyte, x10³/µl, mean±SD | 0.96±1.42 | 0.88±0.50 | 0.696 | 0.83±0.58 | 0.87±0.50 | 0.769 |
| CRP, mg/dL, mean±SD | 15.13±8.69 | 14.66±11.18 | 0.784 | 14.77±9.44 | 14.66±11.18 | 0.961 |
| Procalcitonin, ng/mL, mean±SD | 5.72±20.69 | 16.06±39.90 | 0.046 | 9.94±33.39 | 14.82±38.54 | 0.551 |
| Severity score | | | | | | |
| CURB-65, mean±SD | 2.79±0.97 | 3.23±0.87 | 0.014 | 2.95±0.86 | 3.23±0.87 | 0.154 |
| PSI, mean±SD | 162.81±25.82 | 170.79±30.00 | 0.111 | 165.05±22.23 | 170.79±30.00 | 0.340 |
| Treatment | | | | | | |
| Antiviral agents[h], n (%) | 108 (92.3) | 37 (94.9) | 0.588 | 35 (89.7) | 36 (92.3) | 0.692 |
| Corticosteroid[‡], n (%) | 95 (81.2) | 34 (87.2) | 0.392 | 27 (69.2) | 34 (87.2) | 0.100 |
| Antibiotics[§], n (%) | 100 (85.5) | 34 (87.2) | 0.791 | 34 (87.2) | 33 (84.6) | 0.745 |
| Clinical outcomes | | | | | | |
| Length of hospital stay, days, mean±SD | 16.62±15.31 | 15.77±15.87 | 0.765 | 13.44±11.87 | 15.77±15.87 | 0.464 |
| Mortality, n (%) | 34 (29.1) | 21 (53.8) | 0.009 | 9 (23.1) | 21 (53.8) | 0.005 |
| Bacteremia, n (%) | 15 (12.8) | 9 (23.1) | 0.124 | 6 (15.4) | 9 (23.1) | 0.389 |

Abbreviations: SD, standard deviation; WBC, white blood cell; CRP, C-reactive protein; PSI, pneumonia severity index.

[h] Antiviral agents such as remdesivir, molnupiravir, nirmatrelvir/ritonavir, and others used in the treatment of COVID-19.

[‡] Refers to all corticosteroids, including dexamethasone and methylprednisolone, used in the treatment of COVID-19.

[§] Refers to empirical or targeted antibiotic therapy for confirmed or suspected bacterial infection.

## Discussion

The prevalence of bacterial co-infection in patients with COVID-19 was reported to range from 5.6% to 19% across various studies [3,8–11]. Regardless of the specific prevalence rates, several studies have consistently demonstrated that bacterial co-infection in COVID-19 patients is significantly associated with adverse outcomes, including increased

**Table 3. Risk factors for mortality in patients with COVID-19.**

| | Without propensity score matching | | | | | With propensity score matching | | | | |
|---|---|---|---|---|---|---|---|---|---|---|
| | Univariable analysis | | Multivariable analysis | | | Univariable analysis | | Multivariable analysis | | |
| Variables | OR | *p*-value | OR | 95% CI | *p*-value | OR | *p*-value | OR | 95% CI | *p*-value |
| Sex, male | 1.580 | 0.149 | | | | 1.685 | 0.237 | | | |
| Age | 1.024 | 0.052 | | | | 1.020 | 0.271 | | | |
| Pneumococcal positivity[†] | 2.171 | 0.014 | | | | 2.047 | 0.078 | | | |
| Hypertension | 1.575 | 0.117 | | | | 1.231 | 0.610 | | | |
| Diabetes | 1.228 | 0.483 | | | | 1.142 | 0.737 | | | |
| Chronic lung disease | 0.818 | 0.515 | | | | 1.187 | 0.683 | | | |
| Chronic heart disease | 1.891 | 0.034 | | | | 2.382 | 0.030 | | | |
| Chronic kidney disease | 1.562 | 0.164 | | | | 1.097 | 0.829 | | | |
| Chronic liver disease | 2.875 | 0.124 | | | | 1.368 | 0.801 | | | |
| Solid tumor | 1.100 | 0.777 | | | | 1.022 | 0.968 | | | |
| CVA | 1.053 | 0.875 | | | | 0.828 | 0.674 | | | |
| Hematologic disorder | 1.525 | 0.264 | | | | 0.902 | 0.930 | | | |
| WBC | 1.000 | 0.068 | | | | 1.000 | 0.154 | | | |
| Neutrophil | 1.000 | 0.029 | | | | 1.000 | 0.104 | | | |
| Lymphocyte | 1.000 | 0.282 | | | | 0.999 | 0.033 | | | |
| CRP | 1.060 | <0.001 | | | | 1.041 | 0.049 | | | |
| Procalcitonin | 0.998 | 0.772 | | | | 0.997 | 0.687 | | | |
| CURB-65 | 2.614 | <0.001 | | | | 2.774 | <0.001 | | | |
| PSI | 1.029 | <0.001 | 1.028 | 1.017–1.039 | <0.001 | 1.027 | <0.001 | 1.025 | 1.010–1.039 | 0.001 |
| HFNO | 5.415 | <0.001 | 6.524 | 2.892–14.718 | <0.001 | 9.833 | <0.001 | 9.173 | 2.720–30.933 | <0.001 |
| IMV | 9.150 | <0.001 | 9.994 | 4.423–22.583 | <0.001 | 4.986 | 0.001 | 3.601 | 1.084–11.963 | 0.036 |
| Secondary bacteremia | 8.008 | <0.001 | | | | 5.674 | 0.001 | 2.882 | 0.848–9.796 | 0.090 |
| Antiviral agents[h] | 6.000 | 0.016 | | | | 3.094 | 0.148 | | | |
| Corticosteroids[‡] | 8.785 | <0.001 | | | | 5.418 | 0.008 | | | |
| Antibiotics[§] | 7.219 | 0.001 | | | | 3.048 | 0.035 | | | |

Abbreviations: OR, odds ratio; CI, confidence interval; CVA, cerebrovascular accident; WBC, white blood cells; CRP, C-reactive protein; PSI, pneumonia severity index; HFNO, high flow nasal oxygen; IMV, invasive mechanical ventilation.

[†] Positive result of pneumococcal urinary antigen test or multiplex polymerase chain reaction from sputum samples.

[h] Antiviral agents such as remdesivir, molnupiravir, nirmatrelvir/ritonavir, and others used in the treatment of COVID-19.

[‡] Refers to all corticosteroids, including dexamethasone and methylprednisolone, used in the treatment of COVID-19.

[§] Refers to empirical or targeted antibiotic therapy for confirmed or suspected bacterial infection.

mortality [3,4,12]. Bacterial co-infections in COVID-19 have been attributed to a variety of pathogens, including *Haemophilus influenzae*, *Staphylococcus aureus*, *Streptococcus pneumoniae*, and *Klebsiella pneumoniae* [5].

*S. pneumoniae*, one of the most common causative organisms of CAP, was known to interact with respiratory viruses, including SARS-CoV-2, to induce lower respiratory tract infections [6,7,13,14]. Pneumococcal colonization occurs in approximately 5–20% of healthy adults, predominantly in the upper respiratory tract [15]. This colonization may progress to invasive disease under conditions involving compromised host immunity, such as advanced age, chronic lung disease, immunodeficiency, or co-infection with respiratory viruses—including influenza, respiratory syncytial virus, rhinovirus, and SARS-CoV-2 [6,15,16]. A previous study demonstrated that pneumococcal colonization in COVID-19 patients was associated with reduced salivary IgA (immunoglobulin A) levels specific to SARS-CoV-2 antigens, potentially impairing mucosal

**Table 4. Risk factors for mortality in severe COVID-19 patients (PSI > 130 or CURB-65 ≥ 3).**

| | Without propensity score matching | | | | | With propensity score matching | | | | |
| | Univariable analysis | | Multivariable analysis | | | Univariable analysis | | Multivariable analysis | | |
| Variables | OR | *p*-value | OR | 95% CI | *p*-value | OR | *p*-value | OR | 95% CI | *p*-value |
|---|---|---|---|---|---|---|---|---|---|---|
| Sex, male | 1.365 | 0.419 | | | | 1.353 | 0.573 | | | |
| Age | 0.982 | 0.333 | | | | 0.436 | 0.977 | | | |
| Pneumococcal positivity† | 2.848 | 0.006 | 3.286 | 1.299–8.308 | 0.012 | 3.889 | 0.006 | 4.050 | 1.285–12.765 | 0.017 |
| Hypertension | 1.068 | 0.848 | | | | 0.824 | 0.698 | | | |
| Diabetes | 0.707 | 0.308 | | | | 0.525 | 0.172 | | | |
| Chronic lung disease | 0.774 | 0.473 | | | | 1.100 | 0.848 | | | |
| Chronic heart disease | 1.115 | 0.753 | | | | 1.500 | 0.390 | | | |
| Chronic kidney disease | 0.882 | 0.726 | | | | 0.551 | 0.227 | | | |
| Chronic liver disease | 3.882 | 0.125 | | | | 0.793 | 0.853 | | | |
| Malignancy | 1.038 | 0.923 | | | | 1.464 | 0.534 | | | |
| CVA | 0.626 | 0.193 | | | | 0.506 | 0.167 | 0.372 | 0.114–1.216 | 0.102 |
| Hematological disorder | 2.984 | 0.037 | 4.998 | 1.456–17.157 | 0.011 | NA | >1.00 | | | |
| WBC | 1.000 | 0.315 | | | | 1.000 | 0.730 | | | |
| Neutrophil count | 1.000 | 0.389 | | | | 1.000 | 0.759 | | | |
| Lymphocyte count | 1.000 | 0.795 | | | | 0.999 | 0.156 | | | |
| CRP | 1.021 | 0.250 | | | | 1.005 | 0.832 | | | |
| Procalcitonin | 0.995 | 0.483 | | | | 0.994 | 0.465 | | | |
| CURB-65 score | 2.267 | <0.001 | | | | 2.530 | 0.004 | | | |
| PSI score | 1.028 | <0.001 | 1.031 | 1.015–1.047 | <0.001 | 1.030 | 0.003 | 1.028 | 1.005–1.051 | 0.018 |
| HFNO | 2.662 | 0.005 | 4.530 | 1.812–11.330 | 0.001 | 6.429 | 0.001 | 6.510 | 1.847–22.944 | 0.004 |
| IMV | 4.162 | <0.001 | 7.537 | 2.961–19.186 | <0.001 | 2.143 | 0.170 | | | |
| Secondary bacteremia | 6.008 | <0.001 | | | | 4.300 | 0.017 | | | |
| Antiviral agentsʰ | 2.592 | 0.234 | | | | 1.628 | 0.576 | | | |
| Corticosteroid‡ | 3.760 | 0.020 | | | | 3.706 | 0.056 | | | |
| Antibiotics§ | 4.016 | 0.031 | | | | 1.800 | 0.415 | | | |

Abbreviations: OR, odds ratio; CI, confidence interval; CVA, cerebrovascular accident; WBC, white blood cells; CRP, C-reactive protein; PSI, pneumonia severity index; HFNO, high flow nasal oxygen; IMV, invasive mechanical ventilation.

† Positive result of pneumococcal urinary antigen test or multiplex polymerase chain reaction from sputum samples.

ʰ Antiviral agents such as remdesivir, molnupiravir, nirmatrelvir/ritonavir, and others used in the treatment of COVID-19.

‡ Refers to all corticosteroids, including dexamethasone and methylprednisolone, used in the treatment of COVID-19.

§ Refers to empirical or targeted antibiotic therapy for confirmed or suspected bacterial infection.

immunity. This immunological alteration correlated with higher hospital readmission rates and mortality [6]. These findings are consistent with our study, which observed higher mortality rates among patients with pneumococcal positivity.

In our study, pneumococcal positivity was identified as an independent risk factor for increased mortality among patients with severe COVID-19 compared to those without pneumococcal positivity. This finding likely reflects a greater probability of true pneumococcal co-infection among patients with more severe disease. Consistent with this observation, previous studies have reported increased rates of bacterial co-infection in critically ill patients. A meta-analysis published in 2020 identified bacterial co-infection in 4.9% of COVID-19 patients at the time of hospital admission, rising to 16.0%

among those admitted to the ICU [17]. Similarly, another meta-analysis reported a bacterial co-infection rate of 7% in hospitalized patients with COVID-19, increasing to 14% in ICU settings [18]. Furthermore, the association between bacterial co-infection and poor prognosis is supported by the findings of Patton et al., who reported a mortality rate of 24% among COVID-19 patients with bacterial co-infection—substantially higher than the 5.9% mortality observed in patients with community-acquired bacteremia before the pandemic [19].

We assessed disease severity using the PSI and CURB-65 scoring systems. Although these tools were originally developed for bacterial pneumonia, they have demonstrated strong predictive value for 30-day mortality in patients with COVID-19. Prior studies reported area under the receiver operating characteristic curve (AUC) values of 0.83 and 0.91 for PSI and 0.78 and 0.88 for CURB-65, respectively [20,21], indicating excellent discriminatory performance.

In our study, pneumococcal positivity was defined by either a positive UAT or multiplex PCR result. In clinical practice, timely and accurate identification of the causative pathogen is essential for guiding antimicrobial therapy and improving patient outcomes. However, conventional diagnostic methods such as blood and sputum cultures often fail to identify the pathogen, with detection rates as low as 30–50% in CAP [22–24]. To improve diagnostic yield, adjunctive methods such as UAT and multiplex PCR assays have been increasingly employed. While these methods offer high sensitivity, they are limited by the inability to distinguish colonization from true infection and by the potential for false-positive results. This limitation may have led to misclassification, particularly among patients with mild or moderate COVID-19. Nonetheless, previous evidence suggests that both colonization and active infection with *Streptococcus pneumoniae* are associated with adverse clinical outcomes in patients with COVID-19 [3,4,6,12], indicating that the presence of this pathogen is clinically relevant regardless of its pathogenic state. Furthermore, bacterial co-infections have been reported more frequently in patients with severe COVID-19 [17,18], likely due to factors such as immune dysregulation, impaired mucociliary clearance, and epithelial barrier disruption [25]. Consistent with this, our findings demonstrated a significant association between pneumococcal positivity and mortality only in patients with severe disease, supporting the interpretation that these cases more likely reflect true infection rather than incidental colonization.

This study had several limitations. First, as this was a dual-center retrospective study, the limited number of participating hospitals may affect the generalizability of our findings. Regional and international differences in patient populations, COVID-19 prevalence, and healthcare systems may influence clinical outcomes and limit the applicability of our results. Second, although propensity score matching improved covariate balance, it remains prone to residual confounding and selection bias. This may be partly attributable to variable selection in the matching process, which can omit important confounders. Some variables showed increased imbalance, likely due to the wide caliper used to retain all treated patients. Despite subsequent multivariable adjustment, unmeasured confounding cannot be fully excluded. Therefore, the observed associations should be interpreted with caution. Nevertheless, our data provide valuable insights into the clinical relevance of pneumococcal positivity in patients with severe COVID-19. Third, as noted above, we were unable to distinguish colonization from true infection, potentially resulting in an overestimation of the clinical impact of *S. pneumoniae*. However, since diagnostic tests were performed in patients presenting with respiratory symptoms and radiographic evidence of pneumonia, most cases likely represented clinically meaningful disease. Lastly, the final matched cohorts were composed predominantly of elderly patients, which may reflect the clinical characteristics of severe pneumococcal co-infection. However, this limited age range reduces the generalizability of our findings, particularly to younger populations, and may introduce age-related confounding factors such as comorbidities and baseline mortality risk.

## Conclusion

The detection of *Streptococcus pneumoniae* using urinary antigen testing or multiplex PCR was associated with increased mortality among patients with severe COVID-19. These findings highlight the potential clinical importance of pneumococcal positivity in patients with COVID-19.

## Supporting information

**S1 Table. Covariate balance before and after propensity score matching assessed by standardized mean differences (SMDs) comparing COVID-19 patients with and without pneumococcal positivity.**
(DOCX)

**S1 Data. Pneumococcusin COVID19_rawdata_V2.2.**
(XLSX)

## Author contributions

**Conceptualization:** Chang-Seok Yoon, Tae-Ok Kim, Yu-Il Kim, Sung-Chul Lim.

**Data curation:** Chang-Seok Yoon, Ha-Young Park.

**Formal analysis:** Chang-Seok Yoon.

**Funding acquisition:** Tae-Ok Kim.

**Investigation:** Ha-Young Park, Hwa-Kyung Park, Jae-Kyeong Lee, Bo-Gun Kho.

**Methodology:** Sung-Chul Lim.

**Supervision:** Tae-Ok Kim, Hong-Joon Shin, Yong-Soo Kwon, Yu-Il Kim, Sung-Chul Lim.

**Writing – original draft:** Chang-Seok Yoon.

**Writing – review & editing:** Tae-Ok Kim, Ha-Young Park, Hwa-Kyung Park, Jae-Kyeong Lee, Bo-Gun Kho, Hong-Joon Shin, Yong-Soo Kwon, Yu-Il Kim, Sung-Chul Lim.

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
