## [Decision Letter · Decision Letter 0]

26 Jun 2025

The influence of pneumococcal positivity on clinical outcomes among patients hospitalized with COVID-19: A retrospective cohort study

PLOS ONE

Dear Dr. Kim,

Thank you for submitting your manuscript to PLOS ONE. After careful consideration, we feel that it has merit but does not fully meet PLOS ONE’s publication criteria as it currently stands. Therefore, we invite you to submit a revised version of the manuscript that addresses the points raised during the review process.

We look forward to receiving your revised manuscript.

Kind regards,

Clinton Moodley, Ph.D.

Academic Editor

PLOS ONE

Journal Requirements:

For additional information about PLOS ONE ethical requirements for human subjects research, please refer to http://journals.plos.org/plosone/s/submission-guidelines#loc-human-subjects-research .

“This research was supported by the National Research Foundation of Korea (2022R1F1A1069623) and the Chonnam National University Hospital Biomedical Research Institute (BCRI22033).”

Additional Editor Comments:

Dear authors,

Your manuscript has been peer reviewed by two independent reviewers, and both have highlighted major concerns. We invite you to address these comments and submit a revised manuscript for further consideration.

Reviewers' comments:

Reviewer's Responses to Questions

**Comments to the Author**

1. Is the manuscript technically sound, and do the data support the conclusions?

Reviewer #1: Yes

Reviewer #2: Partly

2. Has the statistical analysis been performed appropriately and rigorously?

Reviewer #1: Yes

Reviewer #2: I Don't Know

3. Have the authors made all data underlying the findings in their manuscript fully available?

Reviewer #1: No

Reviewer #2: No

4. Is the manuscript presented in an intelligible fashion and written in standard English?

Reviewer #1: Yes

Reviewer #2: Yes

Reviewer #1: Clarify the distinction between colonization vs. true infection:

The definition of “pneumococcal positivity” includes UAT and PCR, which do not differentiate between colonization and active infection.

Please elaborate in the Discussion how this limitation may have affected the results, and justify your interpretation that these findings likely reflect clinically significant infection.

Overstatement in Abstract and Conclusion:

Statements such as “was independently associated with increased mortality” may imply causality.

Please revise to state that pneumococcal positivity was “associated with” increased mortality, particularly in severe COVID-19 patients.

Generalizability and Study Limitations:

The study is limited to two hospitals in South Korea. Please emphasize this and potential healthcare system differences in the Discussion.

Clearly state that residual confounding is possible, even after PSM and multivariable adjustments.

Details on Propensity Score Matching:

Include a table of pre- and post-matching covariate balance (standardized mean differences).

Confirm whether caliper width and matching ratio were pre-specified or exploratory.

Data Availability Statement:

Please revise your data availability to comply with PLOS policy. If data are not publicly available, the contact method, specific restrictions, and IRB process should be transparently described.

Reviewer #2: 1. Formatting corrections: Lines 142 and 146 contain formatting inconsistencies regarding Streptococcus pneumoniae. These should be corrected to reflect microbial convention (italicized with the genus capitalized).

2. Inclusion criteria and age representation: Although the study's inclusion criteria specify patients aged over 18 years, the reported mean age of final group of participants analysed is above 70 (e.g., Age (yrs), mean ± SD: 73.16 ± 13.75 and 76.14 ± 13.38; for the PSM group: 75.97 ± 11.74 and 76.14 ± 13.38, p-values 0.121 and 0.93). This raises questions about the absence of younger adults in the final dataset used for analysis. Given that COVID-19 affects all age groups and although community-acquired pneumonia (CAP) disproportionately impacts the elderly, the exclusion of individuals below 70 years is not explained in the manuscript. Moreover, pneumococcal carriage tends to be higher in individuals aged 65 and younger, which challenges the rationale for including only pneumococcus-positive patients aged above 75 (was this the only age group that fitted the matching (PSM)?). The limited age range of patients in the final matched group limits generalizability of results. It further introduces confounding variables such as age-associated comorbidities and inherent mortality risk. It is therefore recommended that the authors mention/discuss this age limitation and its implications for interpretation.

3. Data availability and transparency: While ethical considerations are paramount, the manuscript lacks a clear explanation as to why the deidentified raw data cannot be shared. The authors state they themselved did not access patient identifiers, implying minimal risk to breaking confidentiality. Making deidentified datasets available would allow for independent validation (and confirmation of statistical robustness and workflow) and may help clarify the absence of younger age groups. If published, those wishing to critically appraise the study should be able to access such data upon request.

In summary, although the manuscript requires only minor revisions in formatting and generally reads well, the age-related limitation (point 2) and the absence of transparent data sharing (point 3) are major concerns. The authors may wish to consider revising the manuscript title to reflect the age-specific focus—possibly including the term geriatric—given the lack of data representation from younger cohorts.

Statistical note: The statistical approach appears sound; however, the fact that all matched, included participants were above 70, despite higher pneumococcal carriage rates in younger individuals (as per published data in general), may bias the conclusions. This cohort inherently has a higher mortality risk, which could skew outcome interpretation.

**Do you want your identity to be public for this peer review?** For information about this choice, including consent withdrawal, please see our Privacy Policy

Reviewer #1: No

Reviewer #2: No

---

## [Author Response · Author response to Decision Letter 1]

11 Jul 2025

Review 1

1.Clarify the distinction between colonization vs. true infection: The definition of “pneumococcal positivity” includes UAT and PCR, which do not differentiate between colonization and active infection. Please elaborate in the Discussion how this limitation may have affected the results, and justify your interpretation that these findings likely reflect clinically significant infection.

Thank you for this insightful comment. We agree that the use of UAT and PCR cannot definitively distinguish between colonization and true infection, which constitutes a potential limitation of our study. The interpretation of pneumococcal positivity must therefore be contextualized according to the clinical setting and disease severity. As mentioned in the manuscript and supported by prior meta-analyses, the prevalence of bacterial co-infection increases with the severity of COVID-19. Several mechanisms have been proposed to explain this, including immune dysregulation, impaired mucociliary clearance, and disruption of epithelial barriers in severe cases. These changes can facilitate the transition of colonizing microbiota into invasive pathogens, thereby increasing the likelihood of active infection in severe COVID-19 patients. Thus, in the context of severe COVID-19, pneumococcal positivity is more likely to indicate clinically meaningful co-infection rather than mere colonization. In contrast, in patients with mild to moderate disease, similar test results may more plausibly represent colonization without pathogenic significance. In our study as well, pneumococcal positivity should be interpreted in a context-dependent manner according to disease severity. Importantly, in patients with severe disease, pneumococcal positivity is more likely to indicate active infection, which may in turn contribute to increased mortality.

According to your comment, we added the sentences in 5th paragraph of discussion. Follow as; This limitation may have led to misclassification, particularly among patients with mild or moderate COVID-19. Nonetheless, previous evidence suggests that both colonization and active infection with Streptococcus pneumoniae are associated with adverse clinical outcomes in patients with COVID-19, indicating that the presence of this pathogen is clinically relevant regardless of its pathogenic state. Furthermore, bacterial co-infections have been reported more frequently in patients with severe COVID-19, likely due to factors such as immune dysregulation, impaired mucociliary clearance, and epithelial barrier disruption. Consistent with this, our findings demonstrated a significant association between pneumococcal positivity and mortality only in patients with severe disease, supporting the interpretation that these cases more likely reflect true infection rather than incidental colonization.

2. Overstatement in Abstract and Conclusion:

-Statements such as “was independently associated with increased mortality” may imply causality. Please revise to state that pneumococcal positivity was “associated with” increased mortality, particularly in severe COVID-19 patients.

Thank you for your insightful comment. In accordance with your suggestion, we have revised the Abstract and Introduction to remove the term “independently” in order to avoid implying causality. The revised wording now emphasizes that pneumococcal positivity was associated with, rather than causally linked to, increased mortality—particularly among patients with severe COVID-19. However, we retained the term “independently associated” when reporting multivariable regression results, as it reflects the statistical adjustment for potential confounders.

3.Generalizability and Study Limitations:

-The study is limited to two hospitals in South Korea. Please emphasize this and potential healthcare system differences in the Discussion.

Thank you for your insightful comment. We agree that the limited number of participating hospitals may affect the generalizability of our findings. As noted, the two hospitals included in this study are both tertiary care centers affiliated with Chonnam National University Medical School and are located within the same region of South Korea. This geographical concentration introduces a limitation in terms of regional and institutional diversity.

As you rightly pointed out, differences in patient populations, COVID-19 prevalence, and healthcare systems across regions and countries may influence clinical outcomes and model applicability. Therefore, the generalizability of our findings may be limited, and caution is warranted when applying them to other populations or healthcare settings.

According to your comment, we changed the sentences in 6th paragraph of discussion. Follow as; First, as this was a dual-center retrospective study, the limited number of participating hospitals may affect the generalizability of our findings. Regional and international differences in patient populations, COVID-19 prevalence, and healthcare systems may influence clinical outcomes and limit the applicability of our results.

-Clearly state that residual confounding is possible, even after PSM and multivariable adjustments.

Thank you for your important comment. We agree that, despite the use of propensity score matching and multivariable adjustments, the possibility of residual confounding cannot be entirely excluded due to the observational nature of the study and the potential influence of unmeasured or unknown variables. Accordingly, we have revised and expanded the limitations in discussion of the original manuscript to highlight potential selection bias and residual confounding that may arise from the use of propensity score matching (PSM) and multivariable adjustment.

According to your comment, we changed the sentences in 6th paragraph of discussion. Follow as; Second, although propensity score matching improved covariate balance, it remains prone to residual confounding and selection bias. This may be partly attributable to variable selection in the matching process, which can omit important confounders. Some variables showed increased imbalance, likely due to the wide caliper used to retain all treated patients. Despite subsequent multivariable adjustment, unmeasured confounding cannot be fully excluded. Therefore, the observed associations should be interpreted with caution.

4. Details on Propensity Score Matching:

-Include a table of pre- and post-matching covariate balance (standardized mean differences). Confirm whether caliper width and matching ratio were pre-specified or exploratory.

Thank you for your insightful comment. We have included a new Supplementary Table (Table S3) showing the standardized mean differences before and after PSM. We also clarified in the Methods section that the caliper width (0.2) and 1:1 matching ratio were pre-specified based on previous literature and methodological standards.

According to your comment, we changed the sentences in 2nd paragraph of Methods. Follow as; PSM was performed using the nearest-neighbor approach with a 1:1 matching ratio. To maintain the matching ratio and ensure inclusion of all treated patients, the caliper width was set to 0.2 standard deviations of the logit of the propensity score. To confirm the appropriateness of the matching, we evaluated the balance of matched variables between the two groups using standardized mean differences (SMDs) as the primary measure. Overall, covariate balance was relatively well achieved through propensity score matching, with the majority of variables showing SMDs below 0.1, although some imbalance remained in a few variables. (Table S3). In addition, t-tests and chi-square tests were also performed, where appropriate, to provide supplementary comparisons.

Data Availability Statement:

-Please revise your data availability to comply with PLOS policy. If data are not publicly available, the contact method, specific restrictions, and IRB process should be transparently described.

Thank you for raising this important point. We fully acknowledge the value of data transparency and independent validation. The raw data have been provided as Supporting Information. These data include all relevant variables used in the analysis and are available to facilitate independent verification of the results.

According to your comment, we changed the sentences in Data Availability statement. Follow as; To ensure transparency and reproducibility, the raw data underlying the findings of this study have been included in the Supporting Information files.

Reviewer 2

1. Formatting corrections: Lines 142 and 146 contain formatting inconsistencies regarding Streptococcus pneumoniae. These should be corrected to reflect microbial convention (italicized with the genus capitalized).

Thank you very much for your valuable comments. We have corrected all instances of Streptococcus pneumoniae in the manuscript to follow appropriate microbial formatting conventions in lines 142 and 146

2. Inclusion criteria and age representation: Although the study's inclusion criteria specify patients aged over 18 years, the reported mean age of final group of participants analysed is above 70 (e.g., Age (yrs), mean ± SD: 73.16 ± 13.75 and 76.14 ± 13.38; for the PSM group: 75.97 ± 11.74 and 76.14 ± 13.38, p-values 0.121 and 0.93). This raises questions about the absence of younger adults in the final dataset used for analysis. Given that COVID-19 affects all age groups and although community-acquired pneumonia (CAP) disproportionately impacts the elderly, the exclusion of individuals below 70 years is not explained in the manuscript. Moreover, pneumococcal carriage tends to be higher in individuals aged 65 and younger, which challenges the rationale for including only pneumococcus-positive patients aged above 75 (was this the only age group that fitted the matching (PSM)?). The limited age range of patients in the final matched group limits generalizability of results. It further introduces confounding variables such as age-associated comorbidities and inherent mortality risk. It is therefore recommended that the authors mention/discuss this age limitation and its implications for interpretation.

Thank you for your thoughtful and important comment. Although the inclusion criteria permitted enrollment of patients aged over 18 years, the final matched cohorts were predominantly composed of elderly individuals. This reflects the underlying age distribution of hospitalized patients with severe COVID-19 and pneumococcal positivity during the study period. Younger patients were less likely to meet both the disease severity and pathogen positivity criteria, resulting in limited overlap for propensity score matching.

We agree that this age distribution limits the generalizability of our findings and introduces the potential for age-related confounding. In response, we have added a statement in the Discussion acknowledging the restricted age range of the matched cohorts and its implications for the interpretation of our results.

According to your comment, we added the sentences in last paragraph of Discussions. Follow as; The final matched cohorts were composed predominantly of elderly patients, which may reflect the clinical characteristics of severe pneumococcal co-infection. However, this limited age range reduces the generalizability of our findings, particularly to younger populations, and may introduce age-related confounding factors such as comorbidities and baseline mortality risk.

3. Data availability and transparency: While ethical considerations are paramount, the manuscript lacks a clear explanation as to why the deidentified raw data cannot be shared. The authors state they themselved did not access patient identifiers, implying minimal risk to breaking confidentiality. Making deidentified datasets available would allow for independent validation (and confirmation of statistical robustness and workflow) and may help clarify the absence of younger age groups. If published, those wishing to critically appraise the study should be able to access such data upon request.

Thank you for raising this important point. We fully acknowledge the value of data transparency and independent validation. The raw data have been provided as Supporting Information. These data include all relevant variables used in the analysis and are available to facilitate independent verification of the results.

According to your comment, we changed the sentences in Data Availability statement. Follow as; To ensure transparency and reproducibility, the raw data underlying the findings of this study have been included in the Supporting Information files.

---

## [Editor Report · Decision Letter 1]

17 Jul 2025

The influence of pneumococcal positivity on clinical outcomes among patients hospitalized with COVID-19: A retrospective cohort study

PONE-D-25-19958R1

Dear Dr. Kim,

We’re pleased to inform you that your manuscript has been judged scientifically suitable for publication and will be formally accepted for publication once it meets all outstanding technical requirements.

Kind regards,

Clinton Moodley, Ph.D.

Academic Editor

PLOS ONE

Additional Editor Comments (optional):

The authors have addressed the reviewers comments and concerns adequately, and the manuscript more accurately reflects the data and analyses presented.
---

## [Editor Report · Acceptance letter]

PONE-D-25-19958R1

PLOS ONE

Dear Dr. Kim,

I'm pleased to inform you that your manuscript has been deemed suitable for publication in PLOS ONE. Congratulations! Your manuscript is now being handed over to our production team.

Kind regards,

on behalf of

Dr. Clinton Moodley

Academic Editor

PLOS ONE